# Association between Polyphenol Intake and Lipid Profile of Adults and Elders in a Northeastern Brazilian Capital

**DOI:** 10.3390/nu15092174

**Published:** 2023-05-02

**Authors:** Luciana Melo de Farias, Lays Arnaud Rosal Lopes Rodrigues, Layanne Cristina de Carvalho Lavôr, Alessandro de Lima, Suzana Maria Rebêlo Sampaio da Paz, Jânyerson Dannys Pereira da Silva, Karoline de Macêdo Gonçalves Frota, Massimo Lucarini, Alessandra Durazzo, Daniel Dias Rufino Arcanjo, Maria do Carmo de Carvalho e Martins

**Affiliations:** 1Department of Nutrition, Federal University of Piauí, Teresina 64049-550, PI, Brazil; 2Faculty of Gastronomy, Federal Institute of Piauí, Teresina 64019-368, PI, Brazil; 3Faculty of Pharmacy, Uninassau Alliance College, Teresina 49075-470, PI, Brazil; 4CREA-Research Centre for Food and Nutrition, Via Ardeatina, 546, 00178 Rome, Italy; 5Department of Biophysics and Physiology, Federal University of Piauí, Teresina 64049-550, PI, Brazil

**Keywords:** phytochemicals, polyphenols, food intake, lipoproteins, dyslipidemia

## Abstract

This research was aimed at evaluating the relationship between the estimated polyphenol intake and the atherogenic lipid profile in adult and elder residents in the city of Teresina, located in the Northeastern Region of Brazil. This study was a cross-sectional population-based survey with 501 adults and elders, conducted in Teresina, Brazil. Food intake was obtained by 24-h food recall. The estimated polyphenol intake was calculated by multiplying the food consumption data from the recall by the polyphenol content in the foods described in the Phenol-Explorer database. The mean intake of total polyphenols was 1006.53 mg/day. The phenolic acids was the class with the highest intake, followed by the flavonols. Coffee, beans and apples were the main foods contributing to the total polyphenol intake. In the individuals with elevated serum concentrations of total cholesterol and triglycerides, the intake of total polyphenols was significantly higher. The intake of total polyphenols, phenolic acids and lignans was higher in the subjects with dyslipidemia. This article provides, for the first time, data on the intake of the total polyphenol classes and subclasses in the evaluated population and the relationship with the lipid profile. The individuals with a higher intake of total polyphenols had a worse lipid profile, which may be a consequence of an improved diet in those individuals who present with dyslipidemia.

## 1. Introduction

Cardiovascular diseases (CVD) are the leading cause of morbidity and mortality worldwide, accounting for 32% of all deaths globally [1]. The ten most important risk factors for CVD include unhealthy diet, physical inactivity, dyslipidemia, pre-diabetes, hypertension, obesity, population characteristics such as older age, race/ethnicity and gender differences, kidney dysfunction and genetics [2].

Dyslipidaemia, characterized by elevated serum concentrations of total cholesterol (TC), low-density lipoprotein cholesterol (LDL-c), triglycerides (TG) and reduced concentrations of high-density lipoprotein cholesterol (HDL-c), has been associated with an increased risk of CVD. In particular, increased LDL-c directly influences the development of atherosclerosis. In addition, the reduction in HDL-c concentration also substantially influences the atherogenic lipid profile [3]. 

On the other hand, healthy eating is related to CVD risk reduction [4]. Dietary patterns associated with cardiovascular benefits, such as the Mediterranean diet and other diets rich in specific bioactive compounds, are important strategies in reducing dyslipidemia. Among the compounds identified, polyphenols, present in various food sources, such as teas, coffee, cocoa and apples, stand out [5,6].

Dietary polyphenols [7] are natural compounds widely distributed in foods of plant origin. They are chemically characterized by the presence of one or more aromatic rings attached to hydroxyl groups. They can be subdivided into four main classes: flavonoids, phenolic acids, lignans and stilbenes. In turn, flavonoids are subdivided into six main subclasses: flavonoids, flavonols, anthocyanins, flavones, flavanones and isoflavones [8,9].

Epidemiological and intervention studies have revealed the modulatory effect of some specific polyphenol classes, such as flavonoids, lignans and hydro benzoic acids, on serum lipids [10,11]. These studies show that the consumption of strawberries and apples, sources of anthocyanins and catechins, respectively, significantly reduced the LDL-c concentrations [12]. Likewise, catechins present in green tea significantly decreased the LDL-c concentrations, while cocoa, rich in flavonoids, improved the lipid profile [13,14]. Consistent with these data, the intake of total polyphenols was associated with an improved HDL-c profile [15].

These benefits of polyphenols on cardiovascular health are due to their proven vasodilator action, improvement of the lipid profile, reduction of HDL-c oxidation and modulation of apoptotic processes in the vascular endothelium; such effects may be, in part, attributed to antioxidant and anti-inflammatory properties. However, other mechanisms related to their action on the intestinal microbiota are proposed to explain the effects of these phytochemicals on human health [16].

Despite the recognized beneficial role of polyphenols in the health profile of populations [14,17], data on the estimated intake of these compounds in the Latin American populations are scarce. In Brazil, four studies have presented more consistent estimates of polyphenol intake; one conducted with a sample of adults and elders in the city of São Paulo [18], another with the elders in Viçosa, Minas Gerais [19] and two studies analyzed data from the Pesquisa de Orçamentos Familiares (POF) 2008–2009 [20,21]. 

Due to the great diversity of polyphenols identified in foods of plant origin, their contents and the varying amounts that are ingested (which results in large differences in the daily intake between the polyphenol classes in different populations) [22], it is justified to estimate the polyphenol intake in epidemiological studies, as well as their associations with disease risk.

Therefore, the objectives of this study were to describe the intake of polyphenols and identify the main food sources of polyphenols in adults and elderly residents in the city of Teresina-PI, as well as to evaluate the relationship between the subclasses of polyphenols with higher intake and lipid profile in this population.

## 2. Materials and Methods

### 2.1. Study Population

Data from the Health Survey of the Municipality of Teresina (ISAD-PI) conducted from 2018 to 2020 were used. This was a cross-sectional, population-based study, with probability sampling, stratified by conglomerates, and in two stages: urban census sectors and households. The recruitment protocol and methods were detailed in another study [23]. Individuals residing in private households were eligible to participate in the study, with the exception of those with deficiencies or disabilities that would make the research unfeasible.

The calculation of the sample size was carried out by considering the size and stratification of the population of Teresina, according to the age of the individuals for both sexes. The sample used in this study was composed of data from adults and the elderly in the municipality of Teresina, Piauí, living in private homes in the urban area. The definitions of the World Health Organization (WHO) [24] were considered for the categorization of age groups and, thus, adults were considered as individuals between 20 and 59 years and the elderly were considered as individuals aged 60 years or more.

The ISAD-PI was approved by the Ethics Committee of the Federal University of Piauí on 20 March 2018, under the protocol no. 2.552.426.

### 2.2. Data Collection

The collection of information was carried out in the homes of the participants by previously trained interviewers, divided into two stages. In the first stage, socioeconomic and demographic data, lifestyle and anthropometric measurements and blood pressure were collected. The second stage was carried out in a sub-sample, with the purpose of obtaining food consumption data and blood collection (Figure 1).

The sociodemographic information included gender, age and family income. In addition, smoking and alcohol consumption were investigated, as well as the level of physical activity, measured through the International Physical Activity Questionnaire (IPAQ) [25]. The body mass index (BMI) was obtained through the Quetelet equation [26] 

To determine the lipid profile, the lipid fractions evaluated were total cholesterol (TC), low-density cholesterol (LDL-c), high-density cholesterol (HDL-c) and triglycerides (TG). The concentrations of TC, HDL-c and TG were determined by the colorimetric enzymatic method, using Labtest® (Minas Gerais, Brazil) kits, while the LDL-c fraction was calculated according to the Friedwald formula [27].

Dyslipidemias were classified into four types: isolated hypercholesterolemia (LDL-c ≥ 160 mg/dL), isolated hypertriglyceridemia (TG ≥ 150 mg/dL), hyperlipidemia (LDL-c ≥ 160 mg/dL and TG ≥ 150 mg/dL) and low HDL-c (men < 40 mg/dL and women < 50 mg/dL) isolated or associated with increased LDL-c or TG values [28].

Atherogenic indices were also calculated using the TC/HDL-c, LDL-c/HDL-c ratios, Castelli index I and II [29], respectively, and TG/HDL-c, initially proposed by Gaziano et al. [30]. They were considered as a risk for CVD when the TC/HDL-c ratio was ≥ 3.5, the LDL-c/HDL-c > 2.9 and the TG/HDL-c ratio > 3.7.

The food consumption data were obtained by 24-hour food recall (R24h). To structure the collection of the intake data and minimize errors in their collection [31], the multiple-pass method (MPM) interview technique was used; in this method, the interview is guided through five successive steps [32]:Step 1—Quick list;Step 2—Forgotten list;Step 3—Time and occasion;Step 4—Detail and review;Step 5—Final review.

Food standardization and quantification were performed according to the recommendations of Pinheiro et al. [33], which allowed the conversion of home measures into grams (g) or milliliters (mL). For the culinary preparations, the publications of Fisberg et al. [34], as well as regional and national cookbooks and websites, were also consulted for the identification of individual ingredients.

The food consumption information, generated from the R24h, was converted into energy values (kcal) using the Brazilian Table of Food Composition (TACO) [35]. Furthermore, these data were converted into estimated polyphenol values.

### 2.3. Estimated Intake of Polyphenols

For the estimation of polyphenol intake, the Phenol-Explorer database www.phenol-explorer.eu (accessed on 4 January 2022) was used [36,37]. Phenol-Explorer is the first comprehensive open-access web-database dedicated to polyphenols, containing data and information about the major classes of polyphenols, including flavonoids, phenolic acids, stilbenes, lignans and a small number of other polyphenols, compiled into a class called other polyphenols [36,37].

The polyphenol data available in Phenol-Explorer were obtained by different analytical methods. The data selected for use in this study were obtained by the High-Performance Liquid Chromatography (HPLC) method, while HPLC after hydrolysis and normal phase HPLC were used for a smaller number of foods such as, cereals, pulses, cocoa, seeds and nuts. The content of the total polyphenol or individual polyphenols of the polyphenol class or subclass were expressed as mg/100 g fresh weight or mg/100 mL.

Polyphenol contents not available in the Phenol-Explorer database, such as some Brazilian regional foods rich in these compounds, were obtained by searching articles in the scientific databases Web of Science and MedLine/Pubmed, according to the method followed by Phenol-Explorer.

For minimally processed foods, which were not in the Phenol-Explorer database, such as fruit juices and vitamins, the polyphenol content was estimated based on the percentages of the ingredients. Likewise, the culinary preparations were calculated according to the ingredients in their recipes. The retention factors, available in the Phenol-Explorer database, were applied to adjust the polyphenol content due to the effect of food processing, cooking and storage.

Individual polyphenol intake was obtained by multiplying the daily intake of each food by the content of each compound. Finally, the total intake of polyphenols was estimated from the sum of the individual polyphenol intakes.

### 2.4. Statistical Analysis

Data analysis was performed using the statistical package Stata^®^ version 14. Data were presented as means, standard deviation, medians and interquartile range (IQR). The distribution of polyphenol intake was analyzed using the Shapiro–Wilk test, and did not follow a normal distribution. The Mann–Whitney or Kruskal–Wallis tests were used to verify the difference in the polyphenol intake between the categories of the study variables. To verify the correlation between the polyphenol intake and lipid profile markers (TC, HDL-c, LDL-c and TG), the Spearman’s correlation test was used. Tests with *p*-values < 0.05 were accepted as statistically significant.

## 3. Results

This estimation of polyphenol intake was performed in a sample of 501 individuals, composed of 73.5% adults and 26.5% elders. The intake of polyphenols was evaluated in 204 foods present in the food consumption bank. In total, 284 polyphenols were found in the foods consumed by the research subjects.

The mean intake of total polyphenols was 1006.53 mg/day, with the phenolic acids comprising the class of highest daily intake among the research subjects (480.3 mg/day), followed by the flavonoids (397.9 mg/day), while the lignans, stilbenes and other polyphenols made up the lowest daily intake classes, at 71.2 mg/day, 0.1 mg/day and 11.4 mg/day, respectively. Among the flavonoids, the highest intakes were the flavanols (349.3 mg/day) and the flavonols (27.1 mg/day), while the hydroxycinnamic acids represented the main intake subclass among the phenolic acids (Table 1).

The analysis of the contribution of different food groups to total polyphenol intake showed that non-alcoholic beverages, seeds and fruits were the major sources of polyphenols with respective mean contributions of 617.7 mg/day, 364.7 mg/day and 41.6 mg/day. However, the cocoa/chocolate, spices and oil groups contributed the least to total polyphenol intake.

The three main food sources of total polyphenols were coffee (54.4%), beans (35.1%) and apples (1.2%). Regarding the class of phenolic acids, coffee (95%) was the food with the highest intake, followed by grapes (0.2%) and beer (0.1%). In this same analysis, beans (86.6%), beer (4.1%) and oranges (2.0%) were the main contributing foods to the intake of flavonoids.

The distribution of total polyphenols intake, their classes and subclasses by life stage is described in Table 2. There was no significant difference regarding the intake of total polyphenols when comparing the adult and elderly groups. However, adults presented the highest intake of dihydrochalcones (0.5 mg/day) compared to the elderly (0.136 mg/day). Similarly, adults also showed a higher intake of hydroxybenzoic acids (3.5 mg/day) and hydroxyphenylacetic acids (2.3 mg/day).

The analyses of the relationship between the total polyphenol intake and sociodemographic characteristics, lifestyle and lipid profile are presented in Table 3. Male gender presented significantly higher total polyphenol intake (1318.60 mg/day) compared to female gender (960.34 mg/day). The intake of total polyphenols was significantly higher in individuals with high serum concentrations of TC and TG, when compared to individuals with adequate concentrations (Table 3).

The correlation coefficients of the intake of the total polyphenols, classes and subclasses with the concentrations of total cholesterol, HDL-c, LDL-c and triglycerides are presented in Table 4. A positive and statistically significant correlation (*p* < 0.05) was observed between the intake of total polyphenols, phenolic acids, hydroxynnamic acids and lignans with total cholesterol and triglycerides. Moreover, the anthocyanins and isoflavonoids group showed a positive correlation with the triglycerides and the lignans with LDL-c. In addition to this finding, there was a positive correlation between the intake of flavones, flavanones and hydroxyphenylacetic acid and HDL-c.

Table 5 presents the description of total polyphenols intake, classes and subclasses, according to the presence of dyslipidemia. The intake of total polyphenols (1070.19 mg/day), phenolic acids (520.19 mg/day) and lignans (73.86 mg/day) was significantly higher in the subjects with dyslipidemia when compared to the subjects without dyslipidemia.

## 4. Discussion

Most studies on polyphenol intake and its associations with clinical outcomes are conducted in Europe, North America and Asia [38]. On the other hand, estimates of polyphenol intake in Brazil are still scarce, especially in some regions, such as the northeast of the country. Although two studies that were developed based on data from POF 2008–2009 have described the estimated intake of polyphenols in the Brazilian population, including data from the Northeastern Region [20,21], no other studies with data from this region were identified.

The intake of total polyphenols described here was higher than that observed in the previously mentioned studies. According to Carnaúba et al. (2021) [20], the intake of total polyphenols found in the Northeastern Region was 198.8 mg/day and 204.0 mg/day in the general population, after calorie adjustment. In a sample from the São Paulo Health Survey (ISA-Capital), the mean intake was also lower (377.5 mg/day) [18]. The highest intake of total polyphenols in Brazilian regions was found among elders in the city of Viçosa, Minas Gerais (1198.6 mg/day) [19]. Probably the highest record in our study is due to the consumption of coffee and beans.

Globally, the highest intake of total and individual polyphenols are described in Europe, especially in the Mediterranean countries. In a cohort conducted in ten European countries, the intake ranged from 584 mg/day to 1,786 mg/day [39]. Additionally, according to the study mentioned, the main contributors to total polyphenol intake in the Mediterranean countries were coffee (36%), fruits (25%) and wine (10%), while in the non-Mediterranean European countries, coffee (41%), tea (17%) and fruits (13%) were the main sources of polyphenols, which represents some of the food groups identified throughout our research.

The intake of total polyphenols estimated here was within the values described in Europe. However, relative to other countries, it was higher than the estimates made in Mexico (536 mg/day–750 mg/day) [40]. On the other hand, it was below the intake values of the elderly Japanese (1497 mg/day) [41] and the Polish (1756.5 mg/day) [42]. In the Japanese study, beverage consumption had a major impact on polyphenol intake, accounting for 79% of the intake. Coffee and green tea were the major sources of polyphenols in this population [41]. In the present study, beverages were also the largest contributors to the quantitative intake of polyphenols (64%).

The main class of polyphenol consumed in the studied population was the phenolic acids; this finding was similar to the study of Nascimento-Sousa et al. (2016) [19], who found an intake of 729.5 mg/day. In the city of Catania, Italy, the phenolic acids were also the largest contributors (663.7 mg/day). However, in the study conducted in the Italian city, the main sources of total polyphenols and phenolic acids were nuts and chestnuts [43].

In this study, as well as in the study conducted in the city of São Paulo [19], coffee was the main contributor to the intake of total polyphenols (70.5%) and phenolic acids (92.3%), but these are the highest percentages found among the Brazilian regions studied. This beverage is also an important source of total polyphenols in the Japanese [41] and North-American [44] diets, making up 43.2% and 35%, respectively.

Coffee is one of the most popular beverages in the world; due to that, it contributes significantly to the intake of polyphenols [45]. Chlorogenic acid is the main antioxidant found in coffee. This polyphenol is formed by the esterification of quinic acid with trans-kinnamic acids, mainly caffeic, feluric or *p*-coumaric acids [46]. Studies show that the intake of chlorogenic acid from coffee is associated with several vascular effects, such as antiatherogenic, antihypertensive and antithrombotic effects. The main mechanisms of chlorogenic acid in inhibiting atherosclerosis include increased synthesis and mRNA expression of PPARγ, LXRα, ABCA1 and ABCG1, as well as transcriptional activity resulting from PPARγ activation [47,48].

In a multicenter cohort conducted in Brazil, moderate coffee consumption (1–3 cups/day) reduced the risk of hypertension by 20%, especially in non-smokers. This effect may be attributed to the presence of hypotensive compounds present in coffee, especially phenolic acids, the most abundant polyphenols in this beverage, which can potentially counterbalance the hypertensive effects of caffeine on blood pressure [49].

In the seed group, beans were the second largest contributor to the intake of total polyphenols and the largest contributor of flavonoids in the population of Teresina. These results confirm the findings described by Nascimento-Sousa et al. (2016) [19], as beans contributed 96.9% to the flavanol intake and 70.1% to the flavonol intake. In Northeastern Brazil, beans are a staple food, especially in low-income populations. In this region, cowpea (*Vigna unguiculata*) is widely cultivated and stands out for its nutritional and phytochemical composition [50,51].

Although the fruit group does not present a quantitatively expressive contribution in the consumption of polyphenols, as observed in other regions [52,53], apples were the third food type that mostly contributed to the intake of total polyphenols. Apples are among the main fruits consumed in the world, containing high levels of polyphenols and standing out for their benefits related to human health. Its main phenolic acids include hydroxybenzoic acids, hydroxycinnamic acids, flavanols, flavonols, anthocyanidins and dihydrochalcones [54].

In the analyses performed, the intake of dihydrochalcones differed significantly between the adults and the elders. However, these values were below the intake estimates observed in a cohort conducted in Europe (11.3 mg/day) [42]. These compounds are found mainly in apples, so they are perfect candidates for biomarkers of intake of apples and their derivatives. The main dihydrochalcones found in apples are floretins in free or glycosidic form and floridzins [54].

In turn, the intake of hydroxyphenylacetic acids in the elders was lower when compared to the value described in the Spanish study Prevention with Mediterranean Diet (0.9 mg/day), but in the adults, the intake was higher [15]. As described in the aforementioned study, the main contributors to the intake in this subclass of phenolic acids were olives, red wine, beer and extra virgin olive oil. In contrast, in the population studied, wine consumption was not observed, but grapes were the major contributor to the intake of hydroxyphenylacetic acids. The differences or variations between the results obtained and those described in these studies may reflect regional/continental/cultural differences. 

Sociodemographic variables such as gender and age significantly influence the amount and the polyphenol intake pattern [55]. In the analysis of polyphenol intake according to gender, men showed a higher intake of total polyphenols. These results differed from those found in other studies conducted with populations with similar characteristics, in which the female gender showed a higher intake of polyphenols and also in which a higher total intake of foods contributed to the polyphenol intake among the male gender, whereas the female gender had a better-quality diet [20,56]. As for the relationship between age and the pattern of polyphenol intake, although an increase in the intake of fruits among an older Brazilian population has been observed [57], in this study the lower intake of dihydrochalcones and hydroxyphenylacetic acids by the elders may be also influenced by socioeconomic factors, such as difficulty in chewing the in natura fruit and a preference for the fruit juice [58]. 

Individuals with a higher intake of total polyphenols had higher concentrations of TC and TG. In the presence of dyslipidemia, as defined by one or more increased lipid fractions, a higher intake of total polyphenols and of the main polyphenol classes was also observed. Furthermore, some positive correlations were found between the total, individual polyphenol intakes and the lipid parameters. However, these relationships were weak, and these results could possibly be attributed to reverse causality, common in cross-sectional studies [59]. 

The analysis of the results in this study indicates that the individuals with alterations in the lipid parameters tend to adopt a healthier diet, with the inclusion of foods rich in polyphenols. In light of that, in a study conducted in Teresina and Picos that assessed dietary quality, it was observed that adult individuals with weight excess and metabolic risk had either a very good or excellent diet, findings that were associated with reverse causality [60]. 

The effect of the polyphenols in the lipid profile is still controversial; there is growing evidence of a beneficial role of these compounds in the decrease in the cardiometabolic risk that is associated with the capacity to reduce LDL-c while HDL-c increases [61,62]. Nevertheless, in a meta-analysis conducted by Cao et al. (2022) [63], resveratrol significantly decreased CT, TG and LDL-C, but did not change the HDL-c concentration. As for the relationship between polyphenol intake and HDL-c, differently, in this study there was a positive correlation between the intake of flavones, flavanones and hydroxyphenylacetic acids with the HDL-c concentration. Similarly, in a study conducted with individuals with metabolic syndrome, associations between the flavonoids and the phenolic acids with HDL-c were also found. [15]

According to Luna-Castilho et al. (2021) [64], not only a healthy HDL-c concentration, but also its adequate functionality is important for maintaining the anti-inflammatory, antiatherogenic and antioxidant properties of this molecule. Polyphenols such as quercetin, resveratrol and curcumin, among others, may increase the HDL-c functionality by improving the cholesterol efflux capacity, due to the increased expression of ABCG1 and ABCA1 transporters. In addition, they increase the expression of paraoxonase 1 and contribute to the reduction of cholesterol ester transfer protein activity.

Although the intake of polyphenols is associated with a decrease in the risk factors for chronic diseases, their positive effects depend not only on the amount ingested, but also on their bioavailability, since there is variation among the different classes. In this sense, it should be noted that, the bioavailability of polyphenols in decreasing order is as follows: phenolic acids, isoflavones, flavonols, catechins, flavanones, proanthocyanidins and anthocyanins. The low bioavailability of polyphenols can be attributed to several factors, including interactions with the food matrix, metabolic processes mediated by the liver, intestine and microbiota and variability in systemic absorption even after repeated administration [65].

The divergences between the studies described here can be explained by methodological differences, especially regarding the food consumption data collection instrument, the database used and the chemical form of aglycone. According to the systematic review conducted by Del Bo et al. [38], the intake of polyphenols is generally carried out by means of a 24-hour food recall (56%) and a food frequency questionnaire (31%). As for the databases, the most commonly used were the United States Department of Agriculture (USDA) database and Phenol- Explorer, either alone or in combination. In addition, others were cited, such as the databases Phytonutrient, Flaviola, Finele, and other databases developed by the authors of the papers evaluated or in search of articles.

Regarding the chemical form of polyphenols in food, which may influence the biological activity, in the present study there was no conversion of glycosidic and esterified forms to aglycones, which may overestimate the intake of polyphenols. However, two Brazilian studies, which also presented divergent results, performed a chemical conversion, i.e., polyphenol intake was calculated as aglycone equivalents, considering only the phenolic part of the molecule. The analysis of the POF 2007–2008 [19] presented values that were much lower than those described by Nascimento-Sousa et al. [18], in the city of Viçosa, Minas Gerais. In contrast, the latter study showed similar results to the present research carried out in the city of Teresina, Brazil.

Cultural habits, food preferences, and population income are also contributing factors to the differences found. Furthermore, the scarcity of data on the polyphenol content and subclasses in regional foods made it difficult to obtain a more sensitive analysis. Therefore, it is important to emphasize the need for further studies on polyphenol content, especially in Brazilian regional fruits and other vegetables, either based on the methodology explored here or on other analysis systems.

## 5. Conclusions

The present study provided, for the first time, a broad description of the dietary intake of the total, individual polyphenols and their contributors in adults and elderly people in the study region. Polyphenol intake was similar to that found in other populations, mainly in Europe, but differed as to the main dietary contributors. This information is useful for identifying the health effects of these compounds. The intake of total polyphenols, phenolic acids and lignans was higher in the subjects with dyslipidemia, so polyphenol intake was not associated with a better lipid profile, with the exception of HDL-c.

## Figures and Tables

**Figure 1 nutrients-15-02174-f001:**
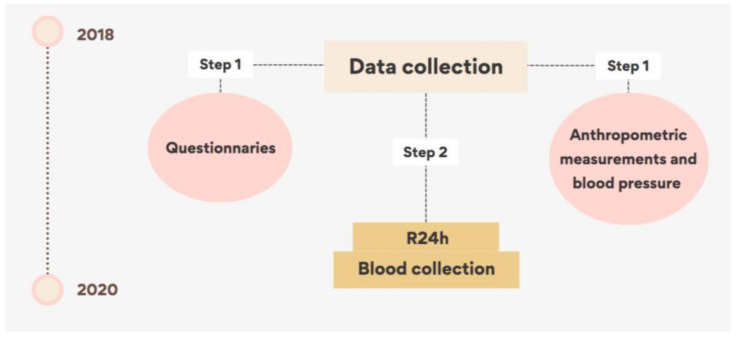
Graphical scheme of data collection.

**Table 1 nutrients-15-02174-t001:** Intake of total polyphenols, classes and subclasses (mg/day), according to food groups and main food contributors.

Classes and Subclasses of Polyphenols	Food Groups										Contribution % of Main Foodstuffs
	Alcoholic Beverages	Non-Alcoholic Beverages	Fruit	Vegetables	Cereals	Cocoa and Chocolate	Seeds	Oils	Seasonings	Total	
	Mean (SD)	Mean (SD)	Mean (SD)	Mean (SD)	Mean (SD)	Mean (SD)	Mean (SD)	Mean (SD)	Mean (SD)	Mean (SD)	
Total polyphenols	25.27 (240.76)	617.70 (603.40)	41.58 (100.83)	14.16 (33.99)	10.48 (33.23)	2.00 (17.01)	364.70 (605.50)	0.79 (8.90)	1.56 (24.62)	1006.53 (837.15)	Coffee (54.4.0%), Beans (35.1%), Apple (1.2%)
Flavonoids	12.29 (116.70)	8.21 (48.92)	27.16 (71.66)	2.81 (6.62)	0.03 (0.27)	1.99 (16.86)	344.90 (589.50)	0.00 (0.04)	0.00 (0.08)	397.90 (628.60)	Beans (86.6%), Beer (4.1%), Orange (2.0%)
Flavones	1.37 (13.18)	0.71 (5.37)	-	0.51 (1.12)	0.01 (0.15)	-	-	0.00 (0.02)	-	2.59 (14.29)	Potato (27.4%), Beer (21.6%), Orange Juice (6.6%)
Flavanols	0.53 (5.06)	0.96 (5.48)	16.56 (48.92)	-	0.00 (0.09)	1.98 (16.81)	324.80 (564.80)	-	0.00 (0.01)	349.30 (586.20)	Beans (93.0%), Banana (0.8%), Chocolate (0.4%)
Flavonols	4.13 (39.79)	0.57 (5.56)	2.12 (21.37)	1.79 (5.03)	-	0.01 (0.08)	18.60 (27.66)	-	0.00 (0.07)	27.21 (52.80)	Beans (67.7%), Beer (29.2%), Grapes (5.1%)
Flavanones	5.66 (53.88)	5.98 (39.03)	8.48 (37.67)	0.50 (1.10)	-	-		0.00 (0.02)		20.62 (75.67)	Orange (39.6%), Beer (22.2%), Orange Juice (14.6%)
Anthocyanins	-	-	-	0.01 (0.19)	0.02 (0.17)	-	1.28 (7.47)	-	-	1.31 (7.49)	Beans (97.7%), Corn (10.7%), Onion (1.1%)
Isoflavonoids	0.61 (5.44)	-	-	-	-	-	0.24 (0.51)	-	-	0.85 (5.45)	Beer (71.8%), Beans (25.9%)
Dihydrochalcones	-	-	0.37 (1.58)	-	-	-	-	-	-	0.37 (1.58)	Apple (100.0%))
Chalcones	0.46 (4.46)	-	-	-	-	-	-	-	-	0.46 (4.46)	Beer (100.0%)
Phenolics acids	1.42 (13.58)	450.40 (442.50)	1.64 (25)	5.05 (14.29)	3.41 (30.05)	0.01 (0.21)	17.69 (32.07)	0.65 (8.89)	0.01 (0.08)	619.63 (3057.40)	Coffee (95.0%), Grapes (0.2%), Beer (0.1%)
Hydroxybenzoic acids	0.80 (7.71)	1.25 (8.54)	-	0.56 (2.11)	0.44 (0.48)	0.01 (0.12)	0.07 (0.79)	0.04 (0.25)	0.003 (0.039)	3.15 (11.59)	Beer (27.3%) Coffee (21.3%) Carrot (12.4%)
Hydroxycinnamic acids	0.62 (5.88)	449.20 (441.80)	1.64 (6.25)	4.49 (13.38)	3.09 (30.63)	0.01 (0.09)	17.62 (31.99)	0.60 (8.80)	0.003 (0.038)	615.90 (3057.30)	Coffee (95.4%), Beans (2.8%), Potatoes (0.4%)
Hydroxyphenylacetic acid	0.05 (0.44)	-	1.75 (14.01)	-	-	-	-	0.00 (0.00)	-	1.79 (14.01)	Grapes (62.2%), Beer (2.2%), Olive Oil (0.1%)
Stilbenes	-	-	0.11 (1.055)	-	-	-	0.00 (0.00)	-	0.002 (0.033)	0.11 (1.06)	Grape (95.3%), Vinegar (0.9%), Peanut (0.1%)
Lignans	0.94 (9.34)	42.81 (42.00)	9.14 (19.61)	5.98 (19.08)	5.57 (6.64)	Traces	6.83 (16.88)	0.02 (0.21)	-	71.23 (52.67)	Coffee (58.4%), Beans (6.8%), Banana (4.7%)
Other polyphenols	7.57 (72.95)	3.27 (3.65)	0.04 (0.15)	0.02 (0.14)	1.65 (8.12)	0.01 (0.14)	0.00 (0.04)	0.10 (0.86)	0.00 (0.03)	11.43 (67.53)	Beer (66.2%), Coffee (25.6%), Noodles (10.4%)

**Table 2 nutrients-15-02174-t002:** Intake of total polyphenols, classes and subclasses (mg/day), according to life cycle.

Classes and Subclasses of Polyphenols	Adults (*n* = 368)		Elderly (*n* = 133)		*p*-Value
	Mean (SD)	Median (IQR)	Mean (SD)	Median (IQR)	
Total polyphenols	1120.01 (904.70)	906.11 (494.11–1532.49)	969.59 (712.02)	870.71 (460.60–140.72)	0.268
Flavonoids	415.99 (679.92)	158.25 (11.83–541.29)	348.95 (473.27)	191.15 (13.71–447.15)	0.860
Flavones	3.070 (16.51)	0 (0–0.80)	1.26 (3.66)	0 (0–0.81)	0.842
Flavanols	364.46 (628.43)	93.69 (1.62–432.12)	308.26 (452.82)	137.69 (7.43–426.68)	0.690
Flavonols	29.95 (59.66)	13.40 (2.34–30.28)	19.63 (24.20)	13.79 (1.25–28.72)	0.339
Flavanones	21.60 (81.52)	0 (0–1.34)	17.91 (56.60)	0 (0–0.97)	0.651
Anthocyanins	1.57 (8.52)	0 (0–0)	0.58 (3.15)	0 (0–0)	0.139
Isoflavonoids	1.073 (6.34)	0.10 (0–0.28)	0.25 (0.65)	0.09 (0–0.21)	0.634
Dihydrochalcones	0.46 (1.72)	0 (0–0)	0.14 (1.10)	0 (0–0)	0.015
Chalcones	0.61 (5.19)	0 (0–0)	0.04 (0.48)	0 (0–0)	0.230
Phenolic acids	489.50 (433.32)	415.51 (185.08–692.44)	454.83 (467.40)	325.29 (128.27–629.29)	0.159
Hydroxybenzoic acids	3.47 (12.53)	0.85 (0.48–1.80)	2.268 (8.46)	0.61 (0.38–0.98)	0.002
Hidroxicinnamic acids	486.03 (433.08)	411.09 (171.00–688.13)	452.56 (465.56)	324.90 (127.77–661.02)	0.177
Hydroxyphenylacetic acids	2.25 (16.09)	0 (0–0)	0.53 (4.62)	0 (0–0)	0.003
Stilbenes	0.14 (1.23)	0 (0–0)	0.003 (0.03)	0 (0–0)	0.408
Lignans	71.78 (52.78)	58.94 (34.35–94.58)	69.71 (52.52)	55.28 (33.98–89.33)	0.660
Other polyphenols	13.46 (78.72)	3.34 (1.65–6.25)	5.95 (10.97)	3.30 (1.09–6.34)	0.619

**Table 3 nutrients-15-02174-t003:** Intake of total polyphenols (mg/day) according to the study variables.

Variables	*n*	Mean (SD)	Median	IQR	*p*-Value
Sex					
Male	167	1318.60 (1053.63)	1078.60	628.63–1695.72	<0.001
Female	334	960.34 (715.12)	801.19	449.33–1322.33	
Age group					
Adult	368	1120.01 (904.70)	906.11	494.11–1532.49	0.268
Elder	133	969.59 (712.02)	870.71	460.60–1340.72	
Income (minimum wages)					
<1	35	1231.74 (898.82)	1075.23	616.50–1438.86	
Between 1 and 2	294	1095.01 (943.68)	845.75	460.60–1473.90	
Between 3 and 4	98	1014.11 (710.98)	915.96	494.11–1460.30	0.808
Between 5 and 9	52	1042.71 (667.76)	990.78	540.61–1383.14	
Between 10 and 20	20	971.01 (625.41)	801.01	455.37–1619.17	
Smoke					
No	387	1106.47 (898.79)	912.68	493.99–1468.96	0.538
Yes	112	988.74 (710.72)	863.08	453.68–1433.99	
Alcoholic beverage					
Never had a drinking habit	180	1052.26 (780.16)	918.50	455.37–1465.21	
Previously did drink, but does not drink currently	118	1099.44 (871.18)	835.82	501.82–1383.86	0.955
Has a drinking habit	201	1093.50 (871.18)	950.30	498.02–1460.30	
BMI					
<25	184	1105.82 (828.08)	902.70	504.17–1498.04	0.442
≥25	317	1065.09 (878.18)	892.29	445.55–1438.86	
Physical activity					
Sedentary	92	964.37 (767.40)	841.18	378.04–1323.94	0.168
Non-sedentary	404	1106.86 (882.17)	911.48	498.90–1492.77	
Diastolic blood pressure (mmHg)					
<80	229	1066.31 (797.39)	929.28	489.57–1431.67	0.853
≥80	267	1092.40 (916.72)	870.71	461.41–1475.82	
Systolic blood pressure (mmHg)					
<120	195	1079.88 (807.65)	941.91	494.11–1461.46	0.710
≥120	301	1080.70 (898.26)	865.60	475.79–1460.30	
Total cholesterol					
<190	183	959.18 (851.01)	705.06	374.51–1350.83	0.022
≥190	96	1080.99 (681.25)	1066.01	522.84–1484.21	
HDL-c					
>40	99	1,068.46 (813.61)	972.68	463.24–1492.77	0.234
≤40	175	955.34 (70,907.39)	788.85	426.16–1329.10	
LDL-c					
<130	205	990.12 (851.77)	720.05	394.16–1406.66	0.149
≥130	74	1031.50 (627.36)	1016.37	499.99–1387.26	
Triglycerides					
<150	142	941.94 (874.21)	705.40	368.98–1240.64	0.020
≥150	137	1,062.41 (707.42)	972.68	499.99–1485.42	
Castelli Index 1 (TC/HDL)					
<3.5	70	999.53 (849.45)	798.20	343.55–1492.77	0.747
≥3.5	204	995.07 (783.43)	595.06	282.49–1022.68	
Castelli Index 2 (LDL/HDL)					
≤2.9	144	1031.47 (887.51)	786.96	414.68–1450.48	0.982
>2.9	227	957.16 (689.73)	872.24	445.55–1329.10	
Triglycerides/HDL					
≤3.7	133	974.55 (862.52)	820.05	379.92–1329.10	0.294
>3.7	141	1016.64 (737.08)	881.17	468.69–1438.86	

BMI: body mass index; (TC), low-density lipoprotein-cholesterol (LDL-c), high-density lipoprotein-cholesterol (HDL-c); TC: total cholesterol; SD: standard deviation; IQR: interquartile range (25% (1st quartile) and 75% (3rd quartile).

**Table 4 nutrients-15-02174-t004:** Correlation between polyphenols intake (mg/day) and lipid profile of adults and elderly.

Variables (*n* = 274)	Total Cholesterol	HDL-c	Triglycerides	LDL-c
	Rho (*p*-Value)	Rho (*p*-Value)	Rho (*p*-Value)	Rho (*p*-Value)
Total polyphenols	0.136 (0.023)	0.044 (0.463)	0.158 (0.008)	0.105 (0.081)
Flavonoids	0.066 (0.296)	0.016 (0.795)	0.009 (0.884)	0.079 (0.217)
Flavones	0.069 (0.251)	0.171 (0.004)	−0.029 (0.623)	0.045 (0.454)
Flavanols	0.071 (0.266)	0.026 (0.685)	−0.003 (0.960)	0.085 (0.183)
Flavonols	0.018 (0.763)	0.035 (0.556)	0.085 (0.160)	0.002 (0.972)
Flavanones	0.074 (0.220)	0.139 (0.020)	−0.057 (0.342)	0.070 (0.246)
Anthocyanins	−0.003 (0.951)	0.021 (0.717)	0.145 (0.015)	−0.080 (0.183)
Isoflavonoids	−0.007 (0.903)	−0.004 (0.945)	0.140 (0.020)	−0.039 (0.517)
Dihydrochalcones	0.040 (0.509)	0.082 (0.175)	−0.054 (0.368)	0.042 (0.489)
Chalcones	0.029 (0.633)	0.014 (0.808)	0.032 (0.591)	0.029 (0.628)
Phenolic acids	0.127 (0.035)	0.032 (0.590)	0.165 (0.006)	0.095 (0.115)
Hydroxybenzoic acids	0.055 (0.359)	0.095 (0.114)	0.037 (0.535)	0.024 (0.687)
Hydroxycinnamic acids	0.126 (0.035)	0.030 (0.619)	0.167 (0.005)	0.095 (0.115)
Hydroxyphenylacetic acids	0.037 (0.535)	0.125 (0.037)	−0.059 (0.322)	0.037 (0.540)
Stilbenes	0.090 (0.133)	0.108 (0.072)	−0.074 (0.217)	0.106 (0.078)
Lignans	0.186 (0.001)	0.104 (0.083)	0.161 (0.007)	0.149 (0.013)
Other polyphenols	0.036 (0.572)	0.024 (0.707)	0.086 (0.176)	0.042 (0.508)

Rho: Spearman’s correlation coefficient.

**Table 5 nutrients-15-02174-t005:** Description of total polyphenol intake and main classes (mg/day), according to the presence of dyslipidemia (adults and elders).

Dyslipidemia (Hypercholesterolaemia, Isolated Hypertriglyceridaemia and Mixed Hyperlipidaemia)					
Classes and Subclasses of Polyphenols	No (*n* = 130)		Yes (*n* = 144)		*p*-Value
	Mean (SD)	Median (IQR)	Mean (SD)	Median (IQR)	
Total polyphenols	914.27 (871.53)	700.28 (365.84–1199.75)	1070.19 (722.94)	987.34 (502.08–1501.42)	0.007
Flavonoids	390.25 (711.20)	176.20 (12.96–439.14)	437.90 (489.49)	101.23 (7.80–449.55)	0.643
Phenolic acids	388.79 (386.73)	327.06 (74.58–571.78)	520.19 (437.34)	452.64 (197.20–754.47)	0.005
Stilbenes	0.04 (0.29)	0 (0–0)	0.001 (0.01)	0 (0–0)	0.233
Lignans	62.81 (51.54)	53.17 (27.74–87.31)	73.86 (47.15)	63.91 (38.29–98.02)	0.012
Other polyphenols	7.13 (18.83)	3.34 (1.27–6.16)	7.93 (29.61)	3.34 (1.59–6.46)	0.552

## Data Availability

Data will be available upon request.

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
