# Peer review of "Association between Polyphenol Intake and Lipid Profile of Adults and Elders in a Northeastern Brazilian Capital"

_nutrients, 2023, doi:10.3390/nu15092174_

Round 1
Reviewer 1 Report
It is well known that the most important risk factor for CVD is unhealthy diet. This Cross-sectional population-based survey aimed to evaluate the relationship between the estimated polyphenol intake and the atherogenic lipid profile in adults and elderly residents in adults and elderly residents in the city of Teresina, located in the Northeast Region of Brazil. The results showed that the individuals with higher intake of total polyphenols had a worse lipid profile, so as to provide a suitable polyphenol intake to those individuals who present dyslipidemia. After reading the manuscript carefully, I can tell you that the information it brings to readers is reasonable and interesting. The work provides, for the first time, data on the intake of total polyphenols classes and subclasses in the evaluated population and relationship with the lipid profile, which will make a major contribution to the field of Lipids, especially the diet improvement in those individuals who present dyslipidemia. In my opinion, there are several minor revisions for this manuscript.
1. English, grammar and typos need to be further improved.
2. Please add some lastest references in the manuscript, especially references in recent two years.
3. In the Discussion section, the authors should compare their own research results with previous reports, especially the different results need to be further discussed.
Author Response
Answer to Reviewer 1
Review Report Form
Open Review
( ) I would not like to sign my review report
(x) I would like to sign my review report
Quality of English Language
( ) English very difficult to understand/incomprehensible
( ) Extensive editing of English language and style required
( ) Moderate English changes required
( ) English language and style are fine/minor spell check required
(x) I am not qualified to assess the quality of English in this paper
|
Yes |
Can be improved |
Must be improved |
Not applicable |
|
|
Does the introduction provide sufficient background and include all relevant references? |
(x) |
( ) |
( ) |
( ) |
|
Are all the cited references relevant to the research? |
(x) |
( ) |
( ) |
( ) |
|
Is the research design appropriate? |
(x) |
( ) |
( ) |
( ) |
|
Are the methods adequately described? |
(x) |
( ) |
( ) |
( ) |
|
Are the results clearly presented? |
( ) |
(x) |
( ) |
( ) |
|
Are the conclusions supported by the results? |
(x) |
( ) |
( ) |
( ) |
Comments and Suggestions for Authors
It is well known that the most important risk factor for CVD is unhealthy diet. This Cross-sectional population-based survey aimed to evaluate the relationship between the estimated polyphenol intake and the atherogenic lipid profile in adults and elderly residents in adults and elderly residents in the city of Teresina, located in the Northeast Region of Brazil. The results showed that the individuals with higher intake of total polyphenols had a worse lipid profile, so as to provide a suitable polyphenol intake to those individuals who present dyslipidemia. After reading the manuscript carefully, I can tell you that the information it brings to readers is reasonable and interesting. The work provides, for the first time, data on the intake of total polyphenols classes and subclasses in the evaluated population and relationship with the lipid profile, which will make a major contribution to the field of Lipids, especially the diet improvement in those individuals who present dyslipidemia. In my opinion, there are several minor revisions for this manuscript.
- English, grammar and typos need to be further improved.
THE MANUSCRIPT WAS ENGLISH PROOFREADED. PLEASE NOTE THE ATTACHED CERTIFICATE.
- Please add some lastest references in the manuscript, especially references in recent two years.
TWO REFERENCES FROM THE LAST TWO YEARS AND ONE FROM 2018 WERE ADDED .
- In the Discussion section, the authors should compare their own research results with previous reports, especially the different results need to be further discussed.
THE AUTHORS HAVE TRIED TO COMPARE THE RESULTS OBTAINED WITH THOSE OF OTHER STUDIES IN MORE DEPTH.
Submission Date
03 March 2023
Date of this review
23 Mar 2023 03:29:48

Reviewer 2 Report
The study by de Farias et al. evaluated the relationship between the estimated polyphenol intake and the atherogenic lipid profile in adults and elderly residents from 2018 to 2020, located in the Northeast Region of Brazil. The paper is well and concisely written, but some things need to be revised.
The number of references in the text should be written before the dot at the end of the sentence.
Please write the date of ethical approval of the research in section 2.1.
Write figures, tables with a capital letter – Figure 1, Table 1.
Correct interviewee to interview in Data collection section.
Please go through the entire text and redo the extra spaces.
This the estimation of polyphenol… - please correct it in Results.
Please write Teresina, 2022 in brackets in Table 1 and others
Please write the numbers in the tables together or separately. Should be uniform throughout the paper - 0.508(1.121) or 0.508 (1.121).
p value write in italics.
Table 2 – should be flavonoids and not flavonoid
Correct Dyslipidemias into dyslipidemia.
p-coumaric acids should be written p-coumaric acids.
References are not written according to the journal's instructions. Please correct all references.
Author Response
Answer to Reviewer 2
Review Report Form
Open Review
(x) I would not like to sign my review report
( ) I would like to sign my review report
Quality of English Language
( ) English very difficult to understand/incomprehensible
( ) Extensive editing of English language and style required
(x) Moderate English changes required
( ) English language and style are fine/minor spell check required
( ) I am not qualified to assess the quality of English in this paper
|
Yes |
Can be improved |
Must be improved |
Not applicable |
|
|
Does the introduction provide sufficient background and include all relevant references? |
(x) |
( ) |
( ) |
( ) |
|
Are all the cited references relevant to the research? |
( ) |
(x) |
( ) |
( ) |
|
Is the research design appropriate? |
( ) |
(x) |
( ) |
( ) |
|
Are the methods adequately described? |
( ) |
(x) |
( ) |
( ) |
|
Are the results clearly presented? |
( ) |
(x) |
( ) |
( ) |
|
Are the conclusions supported by the results? |
( ) |
(x) |
( ) |
( ) |
Comments and Suggestions for Authors
The study by de Farias et al. evaluated the relationship between the estimated polyphenol intake and the atherogenic lipid profile in adults and elderly residents from 2018 to 2020, located in the Northeast Region of Brazil. The paper is well and concisely written, but some things need to be revised.
The number of references in the text should be written before the dot at the end of the sentence.
CORRECTED.
Please write the date of ethical approval of the research in section 2.1.
INCLUDED.
Write figures, tables with a capital letter – Figure 1, Table 1.
CORRECTED.
Correct interviewee to interview in Data collection section.
CORRECTED.
Please go through the entire text and redo the extra spaces.
CORRECTED.
This the estimation of polyphenol… - please correct it in Results.
CORRECTED.
Please write Teresina, 2022 in brackets in Table 1 and others
CORRECTED.
Please write the numbers in the tables together or separately. Should be uniform throughout the paper - 0.508(1.121) or 0.508 (1.121).
CORRECTED.
p value write in italics.
CORRECTED.
Table 2 – should be flavonoids and not flavonoid
CORRECTED.
Correct Dyslipidemias into dyslipidemia.
CORRECTED.
References are not written according to the journal's instructions. Please correct all references.
CORRECTED.
Submission Date
03 March 2023
Date of this review
30 Mar 2023 11:11:59

Reviewer 3 Report
This study evaluated the relationship between polyphenol intake and atherogenic lipid profile in adults and the elderly. The results of this study are important findings for the health function of polyphenols. I believe that the strategies and discussion of the study are reasonable and meet the quality of this journal.
Below I have some suggestions to improve the content of this article.
Comment 1
The study confirmed that the higher the polyphenol intake, the higher the serum total cholesterol and triglyceride concentrations. I felt that there was not enough discussion of this phenomenon. It should be discussed and described in more depth in the discussion section.
Comment 2
The results of this study are important, but somewhat complex and difficult to read. It is recommended that a new Figure (Illustration showing the results of this study) be created and added.
Comment 3
The study adopted data for older adults and adults, as well as data by gender. I felt there was a lack of discussion of how much polyphenol intake could be influenced by age and gender. I recommend that more discussion be added.
Minor comments
I found a few typos even after a short reading. Please recheck the entire text carefully.
Below is some typos that I have found.
Last line of page 3: "Step 4 - Detail and Reveiw" should be "Step 4 - Detail and Review".
2.4.Statistical analysis on page 4: “The Mann-Witney or” should be “The Mann-Whitney or”.
Author Response
Answer to Reviewer 3
Review Report Form
Open Review
(x) I would not like to sign my review report
( ) I would like to sign my review report
Quality of English Language
( ) English very difficult to understand/incomprehensible
( ) Extensive editing of English language and style required
( ) Moderate English changes required
(x) English language and style are fine/minor spell check required
( ) I am not qualified to assess the quality of English in this paper
|
Yes |
Can be improved |
Must be improved |
Not applicable |
|
|
Does the introduction provide sufficient background and include all relevant references? |
( ) |
(x) |
( ) |
( ) |
|
Are all the cited references relevant to the research? |
(x) |
( ) |
( ) |
( ) |
|
Is the research design appropriate? |
(x) |
( ) |
( ) |
( ) |
|
Are the methods adequately described? |
(x) |
( ) |
( ) |
( ) |
|
Are the results clearly presented? |
(x) |
( ) |
( ) |
( ) |
|
Are the conclusions supported by the results? |
(x) |
( ) |
( ) |
( ) |
Comments and Suggestions for Authors
This study evaluated the relationship between polyphenol intake and atherogenic lipid profile in adults and the elderly. The results of this study are important findings for the health function of polyphenols. I believe that the strategies and discussion of the study are reasonable and meet the quality of this journal.
Below I have some suggestions to improve the content of this article.
Comment 1
The study confirmed that the higher the polyphenol intake, the higher the serum total cholesterol and triglyceride concentrations. I felt that there was not enough discussion of this phenomenon. It should be discussed and described in more depth in the discussion section.
THE AUTHORS SOUGHT TO COMPARE THEIR RESULTS WITH THOSE OF OTHER STUDIES IN DEPTH. PLEASE CHECK THE REVISED MANUSCRIPT.
Comment 2
The results of this study are important, but somewhat complex and difficult to read. It is recommended that a new Figure (Illustration showing the results of this study) be created and added.
A GRAPHICAL ABSTRACT WAS INCLUDED IN ORDER TO SUMMARIZE THE STUDY.
Comment 3
The study adopted data for older adults and adults, as well as data by gender. I felt there was a lack of discussion of how much polyphenol intake could be influenced by age and gender. I recommend that more discussion be added.
THE AUTHORS SOUGHT TO DISCUSS THE RELATIONSHIP BETWEEN POLYPHENOL INTAKE AND AGE AND SEX. PLEASE CHECK THE REVISED MANUSCRIPT.
Minor comments
I found a few typos even after a short reading. Please recheck the entire text carefully.
Below is some typos that I have found.
CORRECTED.
Last line of page 3: "Step 4 - Detail and Reveiw" should be "Step 4 - Detail and Review".
CORRECTED.
2.4.Statistical analysis on page 4: “The Mann-Witney or” should be “The Mann-Whitney or”.
CORRECTED.
Submission Date
03 March 2023
Date of this review
28 Mar 2023 07:35:35
